# Fast Adaptive Non-Monotone Submodular Maximization Subject to a Knapsack Constraint

**Georgios Amanatidis**
University of Essex & University of Amsterdam
georgios.amanatidis@essex.ac.uk

**Federico Fusco**
Sapienza University of Rome
fuscof@diag.uniroma1.it

**Philip Lazos**
Sapienza University of Rome
lazos@diag.uniroma1.it

**Stefano Leonardi**
Sapienza University of Rome
leonardi@diag.uniroma1.it

**Rebecca Reiffenhäuser**
Sapienza University of Rome
rebeccar@diag.uniroma1.it

## Abstract

Constrained submodular maximization problems encompass a wide variety of applications, including personalized recommendation, team formation, and revenue maximization via viral marketing. The massive instances occurring in modern-day applications can render existing algorithms prohibitively slow. Moreover, frequently those instances are also inherently stochastic. Focusing on these challenges, we revisit the classic problem of maximizing a (possibly non-monotone) submodular function subject to a knapsack constraint. We present a simple randomized greedy algorithm that achieves a $5.83$ approximation and runs in $O(n \log n)$ time, i.e., at least a factor $n$ faster than other state-of-the-art algorithms. The robustness of our approach allows us to further transfer it to a stochastic version of the problem. There, we obtain a 9-approximation to the best adaptive policy, which is the first constant approximation for non-monotone objectives. Experimental evaluation of our algorithms showcases their improved performance on real and synthetic data.

## 1 Introduction

Constrained submodular maximization is a fundamental problem at the heart of discrete optimization. The reason for this is as simple as it is clear: submodular functions capture the notion of *diminishing returns* present in a wide variety of real-world settings.

Consequently to its striking importance and coinciding NP-hardness [22], extensive research has been conducted on submodular maximization since the seventies (e.g., [16, 44]), with focus lately shifting towards handling the massive datasets emerging in modern applications. With a wide variety of possible constraints, often regarding cardinality, independence in a matroid, or knapsack-type restrictions, the number of applications is vast. To name just a few, there are recent works on feature selection in machine learning [14, 15, 34], influence maximization in viral marketing [3, 33], and data summarization [45, 40, 47]. Many of these applications have *non-monotone* submodular objectives, meaning that adding an element to an existing set might actually decrease its value. Two such examples are discussed in detail in Section 5.

Modern-day applications increasingly force us to face two distinct, but often entangled challenges. First, the massive size of occurring instances fuels a need for very fast algorithms. As the running time is dominated by the *objective function evaluations* (also known as *value oracle calls*), it is typically measured (as in this work) by their number. So, here the goal is to design algorithms requiring an almost linear number of such evaluations. There is extensive research focusing on this issue, be it in the standard algorithmic setting [41], or in streaming [4, 10, 26, 1] and distributed submodular maximization [40, 13]. The second challenge is the inherent uncertainty in problems like sensor placement or revenue maximization, where one does not learn the exact marginal value of an element until it is added to the solution (and thus "paid for"). This, too, has motivated several works on adaptive submodular maximization [27, 28, 30, 43]. Note that even estimating the expected value to a partially unknown objective function can be very costly and this makes the reduction of the number of such calls all the more important.

Knapsack constraints are one of the most natural types of restriction that occurs in real-world problems and are often *hard* budget, time, or size constraints. Other combinatorial constraints like partition matroid constraints, on the other hand, model less stringent requirements, e.g., avoiding too many similar items in the solution. As the soft versions of such constraints can be often hardwired in the objective itself (see the *Video Recommendation* application in Section 5), we do not deal with them directly here. The nearly-linear time requirement, without large constants involved, leaves little room for using sophisticated approaches like continuous greedy methods [24] or enumeration of initial solutions [46]. To further highlight the delicate balance between function evaluations and approximation, it is worth mentioning that, even for the monotone case, the first result combining $O(n \log n)$ oracle calls with an approximation better than 2 is the very recent $\frac{e}{e-1}$-approximation algorithm of Ene and Nguyen [17]. While this is a very elegant theoretical result, the huge constants involved render it unusable in practice.

At the same time, there is a strikingly simple 3-approximation variant of the *modified density greedy* algorithm of Wolsey [49] that deals well with both issues in the *monotone* case: *Sort the items in decreasing order according to their marginal value over cost ratio and pick as many items as possible in that order without violating the constraint. Finally, return the best among this solution and the best single item.*[1] For simplicity, by *modified density greedy* we will refer to *this* algorithm. When combined with lazy evaluations [39], it requires only $O(n \log n)$ value oracle calls and can be adjusted to work well for adaptive submodular maximization [27]. For *non-monotone* objectives, however, the only practical algorithm is the $(10 + \varepsilon)$-approximation FANTOM algorithm of Mirzasoleiman et al. [41] requiring $O(n^2 \log n)$ value oracle calls (see also Remark 1). Moreover, there is no known algorithm for the adaptive setting that can handle anything beyond a cardinality constraint [28].

We aim to tackle both aforementioned challenges for non-monotone submodular maximization under a knapsack constraint, by revisiting the simple algorithmic principle of the modified density greedy algorithm. Our approach is along the lines of recent results on *random greedy* combinatorial algorithms [7, 25], which show that introducing randomness into greedy algorithms can extend their guarantees to the non-monotone case. Here, we give the *first* such algorithm for a knapsack constraint.

## 1.1 Contribution and Outline

The modified density greedy algorithm may produce arbitrarily poor solutions when the objective is non-monotone. In this work we show that introducing some randomization leads to a simple algorithm, SAMPLEGREEDY, that outperforms existing algorithms both in theory and in practice. SAMPLEGREEDY flips a coin before greedily choosing any item in order to decide whether to include it to the solution or ignore it. The algorithmic simplicity of such an approach keeps SAMPLEGREEDY fast, easy to implement, and flexible enough to adjust to other related settings. At the same time the added randomness prevents it from getting trapped in solutions of poor quality.

In particular, in Section 3 we show that SAMPLEGREEDY is a 5.83-approximation algorithm using only $O(n \log n)$ value oracle calls. When all singletons have small value compared to an optimal solution, the approximation factor improves to almost 4. This is the *first* constant-factor approximation algorithm for the non-monotone case using this few queries. The only other algorithm fast enough to

be suitable for large instances is the aforementioned FANTOM [41] which, for a knapsack constraint,[2] achieves an approximation factor of $(10 + \varepsilon)$ with $O(nr\varepsilon^{-1}\log n)$ queries, where $r$ is the size of the largest feasible set and can be as large as $\Theta(n)$. Even if we modify FANTOM to use lazy evaluations, we still improve the query complexity by a logarithmic factor (see Remark 1).

Then we study the problem in the adaptive submodular maximization framework of Golovin and Krause [27] and Gotovos et al. [28], where the stochastic submodular objective is learned as we build the solution and its value depends only on the state of the elements in the evaluated set. For this adaptive variant, we show in Section 4 that a natural adaptation of our algorithm, ADAPTIVEGREEDY, still guarantees a 9-approximation to the *best adaptive policy*. This is not only a relatively small loss given the considerably stronger benchmark, but is in fact the *first* constant approximation known for the problem in this framework. Hence we fill a notable theoretical gap, given that models with incomplete prior information, or those capturing evolving settings, are becoming increasingly important in practice.

From a technical point of view, our algorithm combines the simple principle of always choosing a high-density item with maintaining a careful exploration-exploitation balance, as is the case in many stochastic learning problems. It is therefore directly related to the recent simple randomized greedy approaches for maximizing non-monotone submodular objectives subject to other (i.e., non-knapsack) constraints [7, 10, 25]. However, there are underlying technical difficulties that make the analysis for knapsack constraints significantly more challenging. Every single result in this line of work critically depends on making a random choice in each step, in a way so that "good progress" is consistently made. This is not possible under a knapsack constraint. Instead, we argue globally about the value of the SAMPLEGREEDY output via a comparison with a carefully maintained *almost integral* solution. When it comes to extending this approach to the adaptive non-monotone submodular maximization framework, we crucially use the fact that the algorithm builds the solution iteratively, committing in every step to all the past choices. This is the main technical reason why it is not possible to adjust algorithms with multiple "parallel" runs, like FANTOM, to the adaptive setting.

Our algorithms provably handle well the aforementioned emerging, modern-day challenges, i.e., stochastically evolving objectives and rapidly growing real-world instances. In Section 5 we showcase the fact that our theoretical results indeed translate into applied performance. We focus on two applications that fit within the framework of non-monotone submodular maximization subject to a knapsack constraint, namely *video recommendation* and *influence-and-exploit marketing*. We run experiments on real and synthetic data that indicate that SAMPLEGREEDY consistently performs better than FANTOM while being much faster. For ADAPTIVEGREEDY we highlight the fact that its adaptive behavior results in a significant improvement over non-adaptive alternatives.

## 1.2 Related Work

There is an extensive literature on submodular maximization subject to knapsack or other constraints, going back several decades, see, e.g., [44, 49]. For a *monotone* submodular objective subject to a knapsack constraint there is a deterministic $\frac{e}{e-1}$-approximation algorithm [35, 46] which is tight, unless P = NP [22]. This algorithm has a running time of $O(n^5)$ but there are other, much faster, greedy approaches with weaker approximation guarantees, like the modified density greedy used as a starting point here, Wolsey's 2.8-approximation algorithm [49], and the recent 2-approximation algorithm of Yaroslavtsev et al. [51].

On non-monotone submodular functions Lee et al. [37] provided a 5-approximation algorithm for $k$ knapsack constraints, which was the first constant factor algorithm for the problem. Fadaei et al. [19] building on the approach of Lee et al. [37], reduced this factor to 4. One of the most interesting algorithms for a single knapsack constraint is the 6-approximation algorithm of Gupta et al. [29]. As this is a greedy combinatorial algorithm based on running Sviridenko's algorithm twice, it is often used as a subroutine by other algorithms in the literature, e.g., [13], despite its running time of $O(n^4)$. A number of continuous greedy approaches [24, 36, 9] led to the current best factor of $e$ when a knapsack—or even a general downwards closed—constraint is involved. However, continuous greedy algorithms are impractical for most real-world applications. The fastest such algorithm for our setting is the $(e + \varepsilon)$-approximation algorithm of Chekuri et al. [11] requiring

$O(n^3\varepsilon^{-4}\mathrm{polylog}(n))$ function evaluations. Possibly the only algorithm that is directly comparable to our SAMPLEGREEDY in terms of running time is FANTOM by Mirzasoleiman et al. [41]. FANTOM achieves a $(1+\varepsilon)(p+1)(2p+2\ell+1)/p$-approximation for $\ell$ knapsack constraints and a $p$-system constraint in time $O(nrp\varepsilon^{-1}\log(n))$, where $r$ is the size of the largest feasible solution.

As mentioned above, there is a number of recent results on randomizing simple greedy algorithms so that they work for non-monotone submodular objectives [7, 10, 28, 25, 23]. Our paper extends this line of work, as we are the first to successfully apply this approach for a knapsack constraint.

Golovin and Krause [27] introduced the notions of adaptive monotonicity and submodularity and showed it is possible to achieve guarantees with respect to the optimal adaptive policy that are similar to the guarantees one gets in the standard algorithmic setting with respect to an optimal solution. Our Section 4 fits into this framework as it was generalized by Gotovos et al. [28] for non-monotone objectives. Gotovos et al. [28] showed that a variant of the random greedy algorithm of Buchbinder et al. [7] achieves a $\frac{e}{e-1}$-approximation in the case of a cardinality constraint.

Implicitly related to our quest for few value oracle calls is the recent line of work on the adaptive complexity of submodular maximization that measures the number of sequential rounds of independent value oracle calls needed to obtain a constant factor approximation; see [5, 6, 18, 20, 21] and references therein. For non-monotone functions and a knapsack constraint, Ene et al. [18] give a $O(1)$-approximation algorithm that needs $O(\log^2(n))$ rounds of independent value oracle calls. This, however, does not necessarily translate to a practical query complexity. Using standard arguments for estimating the multilinear extension of the objective and its gradient via sampling [48], the approach of Ene et al. [18] requires $\Omega(n^4)$ oracle queries in the worst case.

## 2 Problem Statement

In this section we formally introduce the problem of submodular maximization with a knapsack constraint in both the standard and the adaptive setting. A function $v : 2^A \to \mathbb{R}^+$ over a set $A$, with $|A| = n$, is *submodular* if the marginal values $v(i \mid S) := v(\{i\} \cup S) - v(S)$ of an element $i \in A \setminus S$ with respect to a set $S \subseteq A$ are diminishing. That is, if $v(i \mid S) \geq v(i \mid T)$ for any $S \subseteq T \subseteq A$ and $i \notin T$. The function $v$ is *non-decreasing* (or simply *monotone*) if $v(S) \leq v(T)$ for any $S \subseteq T \subseteq A$.

In this work we consider general (i.e., not necessarily monotone), normalized (i.e., $v(\emptyset) = 0$), non-negative submodular valuation functions $v$. In Section 3 we assume access to a *value oracle* that returns $v(S)$ when given as input a set $S$. There is a positive cost $c_i$ associated with each element $i \in A$ and a given budget $B$. The goal is to find a subset of $A$ of maximum value among the subsets whose total cost is at most $B$. Formally, we want some $S^* \in \arg\max\{v(S) \mid S \subseteq A, \sum_{i \in S} c_i \leq B\}$. Without loss of generality, we may assume that $c_i \leq B$ for all $i \in A$, since any element with cost exceeding $B$ is not contained in any feasible solution and can be discarded.

We next present the adaptive optimization framework [27, 28]. On a high level, here we do know how the world works and what situations occur with which probability. However, which of those we will be actually dealing with is inferred over time by the bits of information we learn. Along with set $A$, we introduce the *state space* $\Omega$ which is endowed with some probability measure. By $\omega = (\omega_i)_{i \in A} \in \Omega$ we specify the *state* of each element in $A$. The adaptive valuation function $v$ is then defined over $A \times \Omega$; the value over a subset $S \subseteq A$ depends on both the subset and $\omega$. Due to the probability measure over $\Omega$, $v(S, \omega)$ is a random variable. We define $v(S) = \mathbb{E}[v(S, \omega)]$, the expectation being with respect to $\omega$. Like before, the costs $c_i$ are deterministic and known in advance.

For each $\omega \in \Omega$ and $S \subseteq A$, we define the partial realization of state $\omega$ on $S$ as the couple $(S, \omega_{|S})$, where $\omega_{|S} = (\omega_i)_{i \in S}$. It is natural to assume that the true value of a set $S$ does not depend on the whole state, but only on $\omega_{|S}$, i.e., $v(S, \omega) = v(S, \psi)$, for all $\omega, \psi \in \Omega$ such that $\omega_{|S} = \psi_{|S}$. Therefore, sometimes we overload the notation and use $v(S, \omega_{|S})$ instead of $v(S, \omega)$. There is a clear partial ordering on the set of all possible partial realizations: $(S, \omega_{|S}) \subseteq (T, \omega_{|T})$ if $S \subseteq T$ and $\omega_{|T}$ coincides with $\omega_{|S}$ over all the elements of $S$. We are now ready to introduce the concepts of adaptive submodularity and monotonicity. The marginal value of an element $i$ given a partial realization is

$$v(i \mid (S, \omega_{|S})) = \mathbb{E}\left[v(\{i\} \cup S, \omega) - v(S, \omega) \mid \omega_{|S}\right].$$

**Definition 1.** The function $v(\cdot\,,\cdot)$ is *adaptive submodular* if $v(i\,|\,(S,\omega_{|S})) \geq v(i\,|\,(T,\omega_{|T}))$ for all partial realizations $(S,\omega_{|S}) \subseteq (T,\omega_{|T})$ and for any $i \notin T$. Further, $v(\cdot\,,\cdot)$ is *adaptive monotone* if $v(i\,|\,(S,\omega_{|S})) \geq 0$ for all partial realizations $(S,\omega_{|S})$ and for all $i \notin S$.

In Section 4 we assume access to a value oracle that given an element $i$ and a partial realization returns the expected marginal value of $i$. Using the properties of conditional expectation, it is straightforward to show that if $v(\cdot\,,\cdot)$ is adaptive submodular, then its expected value $v(\cdot)$ is submodular. In analogy with [28], we assume $v$ to be state-wise submodular, i.e., $v(\cdot,\omega)$ is a submodular set function for each $\omega \in \Omega$. In this framework it is possible to define *adaptive policies* to maximize $v$. An adaptive policy is a function which associates with every partial realization a distribution on the next element to be added to the solution. The optimal solution to the adaptive submodular maximization problem is to find an adaptive policy that maximizes the expected value while respecting the knapsack constraint (the expectation being taken over $\Omega$ and the randomness of the policy itself).

## 3 The Algorithmic Idea

We present and analyze SAMPLEGREEDY, a randomized 5.83-approximation algorithm for maximizing a submodular function subject to a knapsack constraint. As we mentioned already, SAMPLEGREEDY is based on the modified density greedy algorithm of Wolsey [49]. Since the latter may perform arbitrarily bad for non-monotone objectives, we add a sampling phase, similar to the sampling phase of the Sample Greedy algorithm of Feldman et al. [25].

SAMPLEGREEDY first selects a subset $A'$ of $A$ by independently picking each element with probability $p$. Then it runs Wolsey's algorithm only on $A'$. To formalize this second step, using $v(i)$ as a shorthand for $v(\{i\})$, let $j_1 \in \arg\max_{i \in A'} v(i)/c_i$, and $j_{k+1} \in \arg\max_{i \in A' \setminus \{j_1,\ldots,j_k\}} v(i\,|\,\{j_1,\ldots,j_k\})/c_i$ for $k \geq 1$. If $\ell$ is the largest integer such that $\sum_{i=1}^{\ell} c_{j_i} \leq B$, then $S = \{j_1,\ldots,j_\ell\}$. In the end, the output is the one yielding the largest value between $S$ and an element from $\arg\max_{i \in A'} v(i)$.

Due to space constraints, we defer the pseudo-code of SAMPLEGREEDY as well as the proofs of the statements below to the supplementary material. To facilitate our analysis, however, there we state an equivalent algorithm (in the sense that the two have identical output distributions) where the sampling phase does not proceed the greedy part. Instead, we start with the whole set $A$ but when the algorithm greedily considers an item, it only adds it in the solution with probability $p$. Lines 5-13 of ADAPTIVEGREEDY illustrate how this equivalent greedy solution is built.

**Theorem 1.** *For $p = \sqrt{2} - 1$, SAMPLEGREEDY is a $\left(3 + 2\sqrt{2}\right)$-approximation algorithm.*

A naive implementation of SAMPLEGREEDY needs $\Theta(n^2)$ value oracle calls in the worst case. Indeed, in each iteration all the remaining elements have their marginals updated and for large enough $B$ the greedy solution may contain a constant fraction of $A$. Applying lazy evaluations [39], however, we can cut the number of queries down to $O(n\varepsilon^{-1} \log(n/\varepsilon))$ losing only an $\varepsilon$ in the approximation factor (see also [17]). To achieve this, instead of recomputing all the marginals at every step, we maintain an ordered queue of the elements sorted by their last known *densities* (i.e., their marginal value per cost ratios) and use it to get a *sufficiently good* element to add.

More formally, the lazy implementation of SAMPLEGREEDY maintains the elements in a priority queue in decreasing order of density, which is initialised using the ratios $v(i)/c_i$. At each step we pop the element on top of the queue. If its density with respect to the current solution is within a $1 + \varepsilon$ factor of its old one, then it is picked by the algorithm, otherwise it is reinserted in the queue according to its new density and we pop the next element. Submodularity guarantees that the density of a picked element is at least $1/(1 + \varepsilon)$ of the best density for that step. As soon as an element has been updated $\log(n/\varepsilon)/\varepsilon$ times, we discard it.

**Theorem 2.** *The lazy version of SAMPLEGREEDY achieves an approximation factor of $3 + 2\sqrt{2} + \varepsilon$ using $O(n\varepsilon^{-1} \log(n/\varepsilon))$ value oracle calls.*

Additionally, our analysis implies that SAMPLEGREEDY performs significantly better in the *large instance* scenario, i.e., when the value of the optimal solution is much larger than the value of any single element. While it is not expected to have exact knowledge of the factor $\delta$ in the following proposition, often some estimate is accessible. Especially for massive instances, it is reasonable to assume that $\delta$ is bounded by a very small constant.

**Theorem 3.** *If* $\max_{i \in A} v(i) \le \delta \cdot \mathrm{OPT}$ *for* $\delta \in (0, 1/2)$*, then* SAMPLEGREEDY *with* $p = \frac{1-\delta}{2}$ *is a* $(4 + \varepsilon_\delta)$*-approximation algorithm, where* $\varepsilon_\delta = \frac{4\delta(2-\delta)}{(1-\delta)^2}$.

## 4 Adaptive Submodular Maximization

In this section we modify SAMPLEGREEDY to achieve a good approximation guarantee in the adaptive framework. Recall that the adaptive valuation function $v(\cdot, \cdot)$ depends on the state of the system which is discovered a bit at a time, in an adaptive fashion. Indeed, SAMPLEGREEDY is compatible with this framework and can be applied nearly as it is. We stick to the interpretation of SAMPLEGREEDY discussed right before Theorem 1. That is, there is no initial sampling phase. Instead, we directly begin to choose greedily with respect the density (marginal value with respect to the current solution over cost). Each time we are about to pick an element of $A$, we throw a $p$-biased coin that determines whether we keep or discard the element.

Here the main difference with the greedy part of SAMPLEGREEDY is that the marginals are to be considered with respect to the *partial realization* relative to the current solution. Moreover, since it is not possible to return the largest between $\max_{i \in A} v(i)$ and the result of the greedy exploration, the choice between these two quantities has to be settled before starting the exploration. Formally, at the beginning of the algorithm a $p_0$-biased coin is tossed to decide between the two. The pseudo-code for the resulting algorithm, ADAPTIVEGREEDY, is given below.

---

ADAPTIVEGREEDY

1  Let $r_0 \sim \mathrm{Bernoulli}(p_0)$
2  **if** $r_0 = 1$ **then**
3      $i^* \in \arg\max_{k \in A} v(k)$                        `/* best single item in expectation */`
4      Observe $\omega_{i^*}$ and **return** $v(i^*, \omega_{i^*})$
5  $S = \emptyset, R = B$                              `/* greedy solution and remaining knapsack capacity */`
6  $F = \{k \in A \mid v(k) > 0\}$                           `/* initial set of candidate items */`
7  **while** $F \ne \emptyset$ **do**
8      Let $i \in \arg\max_{k \in F} \frac{v(k \mid (S, \omega_{|S}))}{c_k}$
9      Let $r_i \sim \mathrm{Bernoulli}(p)$                         `/* independent random bit */`
10     **if** $r_i = 1$ **then**
11         Observe $\omega_i :$  $S = S \cup \{i\}, \; R = R - c_i$
12     $A = A \setminus \{i\}, \quad F = \{k \in A \mid v(k \mid (S, \omega_{|S})) > 0 \text{ and } c_k \le R\}$
13 **return** $S, v(S, \omega_{|S})$

---

We state the main result of this section, which is proved in in the supplementary material. Its analysis follows a similar path to the one of Theorem 1, but the underlying randomness of the states creates extra difficulties in some of the steps involved. The two sources of randomness, namely $\Omega$ and the coin tosses $\{r_i\}$, have to be carefully combined and this leads to a small loss in the approximation guarantee. Let $\mathrm{OPT}_\Omega$ denote the expected value of the optimal adaptive policy.

**Theorem 4.** *For $p_0 = 1/3$ and $p = 1/6$,* ADAPTIVEGREEDY *yields a 9-approximation of* $\mathrm{OPT}_\Omega$*, while its* lazy *version achieves a* $(9 + \varepsilon)$*-approximation using $O(n\varepsilon^{-1}\log(n/\varepsilon))$ value oracle calls.*

*Moreover, when $\max_{i \in A} v(i) \le \delta \cdot \mathrm{OPT}_\Omega$ for $\delta \in (0, 1/2)$, then for $p_0 = 0$ and $p = (\sqrt{3 - 2\delta} - 1)/2$,* ADAPTIVEGREEDY *yields a $(4 + 2\sqrt{3} + \varepsilon'_\delta)$-approximation, where $\varepsilon'_\delta \approx \frac{6\delta(2-\delta)}{(1-\delta)^2}$.*

## 5 Experiments

Out of the numerous applications of submodular maximization subject to a knapsack constraint, we evaluate SAMPLEGREEDY and ADAPTIVEGREEDY on two selected examples, using real and synthetic graph topologies. Variants of these have been studied in a similar context; see [41].

A delicate point is tuning the probabilities of acceptance $p$ for improved performance. While the choices of $p$ in Theorems 1 and 4 minimize our analysis of the theoretical worst-case approximation,

there are two factors suggesting that a value closer to 1 works better in real-world applications: the best singleton typically is small and non-monotone objectives are—loosely speaking—not arbitrarily far from monotonicity. Even so, we do not micro-optimize for $p$. Instead, we "guess" a good value for it by running our algorithms 5 times, each with a $p$ chosen uniformly at randomly from $[0.9, 1]$. As the algorithms are extremely fast, this does not affect our ability to handle very large instances. Notice that while it is not realistic to run ADAPTIVEGREEDY multiple times in the adaptive case, this could be seen as a dry run to tune $p$ before trying on actual data.

It is important to note, however, that the theoretically optimal values of $p$ still work well in a number of different scenarios—including all the experiments here—in the sense that the algorithms perform significantly better than their worst-case guarantee. Moreover, the best values of $p$ in practice (which vary among the instances) for *all* the instances in our experiments are strictly less than 1 and typically in $[0.92, 0.97]$. This suggests that randomization helps, even in instances where the deterministic density greedy algorithm would perform well despite the non-monotonicity.

**Video Recommendation:** Suppose we have a large collection $A$ of videos from various categories (represented as possibly intersecting subsets $C_1, \ldots, C_k \subseteq A$) and we want to design a recommendation system. When a user inputs a subset of categories and a target total length $B$, the system should return a set of videos from the selected categories of total duration at most $B$ that maximizes an appropriate objective function. (Of course, instead of time here, we could use costs and a budget constraint.) Each video has a rating and there is some measure of similarity between any two videos. We use a weighted graph on $A$ to model the latter: each edge $\{i, j\}$ between two videos $i$ and $j$ has a weight $w_{ij} \in [0, 1]$ capturing the percentage of their similarity. To pave the way for our $v(\cdot)$, we start from the auxiliary objective $f(S) = \sum_{i \in S} \sum_{j \in A} w_{ij} - \lambda \sum_{i \in S} \sum_{j \in S} w_{ij}$, for some $\lambda \geq 1$ [38, 41]. This is a *maximal marginal relevance* inspired objective [8] that rewards coverage, while penalizing similarity. For $\lambda = 1$, internal similarities are irrelevant and $f$ becomes a cut function. However, one can penalize similarities even more severely as $f$ is submodular for $\lambda \geq 1$ (e.g., Lin and Bilmes [38] use $\lambda = 5$).

In order to mimic the effect of a partition matroid constraint, i.e., the avoidance of many videos from the same category, we may use two parameters $\lambda \geq 1, \mu \geq 0$. While $\lambda$ is as above, $\mu$ puts extra weight on similarities between videos that belong to the same category. That leads to a more general auxiliary objective $g(S) = \sum_{i \in S} \sum_{j \in A} w_{ij} - \sum_{i \in S} \sum_{j \in S} (\lambda + \chi_{ij}\mu) w_{ij}$, where $\chi_{ij}$ is equal to 1 if there exists $\ell$ such that $i, j \in C_\ell$ and 0 otherwise. To interpolate between choosing highly rated videos and videos that represent well the whole collection, here we use the submodular function $v(S) = \alpha \sum_{i \in S} \rho_i + \beta g(S)$ for $\alpha, \beta \geq 0$, where $\rho_i$ is the rating of video $i$. We use $\lambda = 3, \mu = 7$ and set the parameters $\alpha, \beta$ so that the two terms are of comparable size.

We evaluate SAMPLEGREEDY on an instance based on the latest version of the MovieLens dataset [31], which includes 62000 movies, 13816 of which have *both* user-generated tags *and* ratings. We calculate the weights $w_{ij}$ using these tags (with the L2 norm of the pairwise minimum tag vector, see the supplementary material) while the costs are drawn independently from $U(0, 1)$. We compare against the FANTOM algorithm of Mirzasoleiman et al. [41] as it is the only other algorithm with a provable approximation guarantee that runs in reasonable time. Continuous greedy approaches [24] or the repeated greedy of Gupta et al. [29] are prohibitively slow. SAMPLEGREEDY consistently performs better than FANTOM for a wide range of budgets (Figure 1a). Plotting the number of function evaluations against the budget, SAMPLEGREEDY is much faster (Figure 1d) despite the fact that it is run 5 times!

**Remark 1.** The running time of FANTOM for fixed $\varepsilon$ is $O(nr \log n)$, where $r$ is the cardinality of the largest feasible solution. For a knapsack constraint this translates to $O(n^2 \log n)$. To be as fair as possible, we implemented FANTOM using lazy evaluations, which improves the number of evaluations of the objective function to $O(n \log^2 n)$ and is indeed much faster in practice, for the knapsack sizes we consider. Even so, our SAMPLEGREEDY is faster by a factor of $\Omega(\log n)$ which, including the improvement in the constants involved, still makes a huge difference. Note that in both Figures 1d and 1e one can discern the superlinear increase of the function evaluations for FANTOM but not for SAMPLEGREEDY.

**Influence-and-Exploit Marketing:** Consider a seller of a single digital good (i.e., producing extra units of the good comes at no extra cost) and a social network on a set $A$ of potential buyers. Suppose that the buyers influence each other and this is quantified by a weight $w_{ij}$ on each edge $\{i, j\}$ between

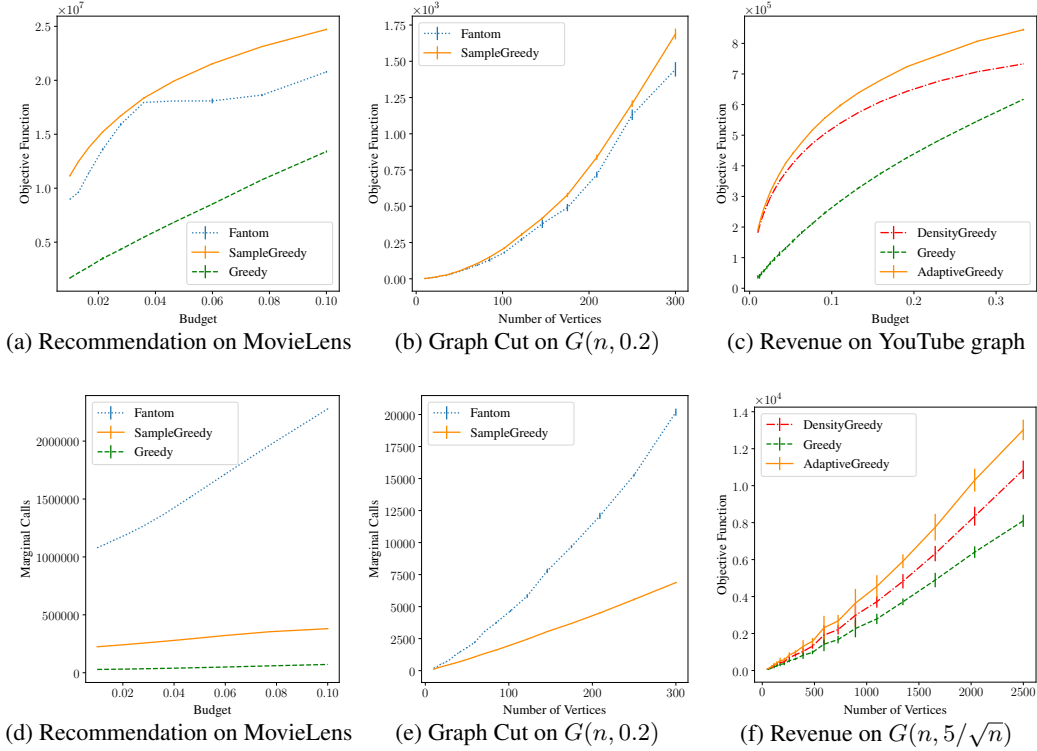

(a) Recommendation on MovieLens    (b) Graph Cut on $G(n, 0.2)$    (c) Revenue on YouTube graph

(d) Recommendation on MovieLens    (e) Graph Cut on $G(n, 0.2)$    (f) Revenue on $G(n, 5/\sqrt{n})$

Figure 1: The four plots on the left compare the performance and the number of function evaluations of SAMPLEGREEDY and FANTOM on the video recommendation problem for the MovieLens dataset (a), (d) and on the maximum weighted cut problem on random graphs (b), (e). Since no $\varepsilon \le 1$ affected the performance of FANTOM noticeably before becoming too computationally expensive, we used $\varepsilon = 1$ to achieve the maximum possible speedup. The plots on the far right illustrate the performance of ADAPTIVEGREEDY (ignoring single item solutions, i.e., $p_0 = 0$) on the influence-and-exploit problem for two distinct topologies: the YouTube graph (c) and random graphs (f). All budgets are shown as fractions of the total cost.

buyers $i$ and $j$. Each buyer's value for the good depends on who owns it within her immediate social circle and how they influence her. A possible revenue-maximizing strategy for the seller is to first give the item for free to a selected set $S$ of influential buyers (influence phase) and then extract revenue by selling to each of the remaining buyers at a price matching their value for the item due to the influential nodes (exploit phase). Here we further assume, similarly to the adaptation of this model by Mirzasoleiman et al. [41], that each buyer comes with a cost of convincing her to advertise the product to her friends. The seller has a budget $B$ and the set $S$ should be such that $\sum_{i \in S} c_i \le B$.

We adopt the generalization of the *Concave Graph Model* of Hartline et al. [32] to non-monotone functions [3]. Each buyer $i \in A$ is associated with a non-negative concave function $f_i$. For any $i \in A$ and any set $S \subset A \setminus \{i\}$ of agents already owning the good, the value of $i$ for it is $v_i(S) = f_i\left(\sum_{j \in S \cup \{i\}} w_{ij}\right)$. The total potential revenue $v(S) = \sum_{i \in A \setminus S} v_i(S)$ that we aim to maximize is a non-monotone submodular function. Besides the theoretical guarantees for influence-and-exploit marketing in the Bayesian setting [32], there are strong experimental evidence of its performance in practice [3]. The problem generalizes naturally to different stochastic versions. We assume that the valuation function of each buyer $i$ is of the form $f_i(x) = a_i\sqrt{x}$ where $a_i$ is drawn independently from a Pareto Type II distribution with $\lambda = 1, \alpha = 2$. We only learn the exact value of a buyer when we give the good for free to someone in her neighborhood.

We evaluate ADAPTIVEGREEDY on an instance based on the YouTube graph [50], containing 1,134,890 vertices. The (known) weights are drawn independently from $U(0, 1)$, and the costs are proportional to the sum of the weights of the incident edges. As ADAPTIVEGREEDY is the first adaptive algorithm for the problem, we compare with non-adaptive alternatives like *Greedy*[3] and

*Density Greedy*[4] for different values of the budget. ADAPTIVEGREEDY outperforms the alternatives by up to $20\%$ (Figure 1c). We observe similar improvements for Erdős-Rényi random graphs of different sizes and edge probability $5/\sqrt{n}$ and a fixed budget of $10\%$ of the total cost (Figure 1f).

**Maximum Weighted Cut:** Beyond the above applications, we would like to compare SAMPLE-GREEDY to FANTOM with respect to both their performance and the number of value oracle calls as $n$ grows. We turn to *weighted cut functions*—one of the most prominent subclasses of non-monotone submodular functions—on dense Erdős–Rényi random graphs with edge probability $0.2$. The weights and the costs are drawn independently and uniformly from $[0, 1]$ and the budget is fixed to $15\%$ of the total cost. Again SAMPLEGREEDY consistently performs better than FANTOM, albeit by $5$–$15\%$ (Figure 1b). In terms of running time, there is a large difference in favor of SAMPLEGREEDY (even for multiple runs), while the superlinear increase for FANTOM is evident (Figure 1e). Since here SAMPLEGREEDY and FANTOM are quite close to each other in terms of performance and Greedy would lie only slightly below the plot of SAMPLEGREEDY, to improve the readability of Figure 1b we have removed Greedy from this comparison.

**Remark 2.** Based on the theoretical query complexities, one would expect the comparison between FANTOM and SAMPLEGREEDY in Figures 1d, 1e to be qualitatively similar to $n \log^2 n$ vs. $n \log n$. However, while FANTOM clearly exhibits a superlinear dependence of the query complexity on the input size, SAMPLEGREEDY does not. The reason for this is that, in practice, lazy evaluations often result in much less than $\log n$ evaluations per element. So, what we see in Figures 1d, 1e is closer to $n \log n$ vs. $n$.

## 6  Discussion

The proposed random greedy method yields a considerable improvement over state-of-the-art algorithms, especially, but not exclusively, regarding the handling of huge instances. With all the subtleties of our work affecting solely our analysis, the algorithm remains strikingly simple and we are confident this will also contribute to its use in practice. Simultaneously, this very simplicity translates into a generality that can be employed to achieve comparably good results for a variety of settings; we demonstrated this in the case of the adaptive submodularity setting.

Specifically, we expect that our approach can be directly utilised to improve the performance and running time of algorithms that now use some variant of the algorithm of Gupta et al. [29]. Such examples include the distributed algorithm of da Ponte Barbosa et al. [13] and the streaming algorithm of Mirzasoleiman et al. [42] in the case of a knapsack constraint. We further suspect that the same algorithmic principle can be applied in the presence of incentives. This would largely improve the current state of the art in budget-feasible mechanism design for non-monotone objectives [12, 2].

A different direction would be to try other greedy algorithms for monotone objectives as a starting point. For instance, the 2-approximation algorithm of Yaroslavtsev et al. [51] could potentially yield a better approximation ratio for the standard algorithmic setting. Unfortunately, it does not seem possible to translate more involved algorithms like that one to the adaptive setting, where one has to commit to all of their past choices.

Finally, a major question here is whether the same high level approach is valid even in the presence of additional combinatorial constraints. In particular, is it possible to achieve similar guarantees as FANTOM for a $p$-system and multiple knapsack constraints using only $O(n \log n)$ value queries?

## Broader Impact

While the performance of our algorithms constitutes a significant improvement over existing methods with regard to speed as well as the handling of uncertain environments, there already exists a vast body of research on the type of problems they can be applied to. Our methods could lead to handling those problems, e.g., summarising training data, recommendation systems, viral marketing, etc., more efficiently. Even though these applications revolutionized many areas of business and society, we do not expect the impact of our research to be substantially different from existing methods.

## Acknowledgments and Disclosure of Funding

This work was supported by the ERC Advanced Grant 788893 AMDROMA "Algorithmic and Mechanism Design Research in Online Markets", the MIUR PRIN project ALGADIMAR "Algorithms, Games, and Digital Markets", and the NWO Veni project VI.Veni.192.153.

The authors have no competing interests to disclose.

## Footnotes

[1] A somewhat more elaborate 2.8-approximation algorithm is given by Wolsey [49]. For completeness, in the supplementary material we show the approximation guarantee of the simplified version we mention here.

[2]FANTOM can handle more general constraints, like a $p$-system constraint and $\ell$ knapsack constraints. Here we refer to its performance and running time when restricted to a single knapsack constraint.

[3]The simple greedy algorithm that in each step picks the element with the largest marginal value.

[4]The deterministic density greedy part of Wolsey's algorithm [49].

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
