[Supplementary Material]

# Fast Adaptive Non-Monotone Submodular Maximization Subject to a Knapsack Constraint
## *Supplementary Material*

**Georgios Amanatidis**
University of Essex & University of Amsterdam
georgios.amanatidis@essex.ac.uk

**Federico Fusco**
Sapienza University of Rome
fuscof@diag.uniroma1.it

**Philip Lazos**
Sapienza University of Rome
lazos@diag.uniroma1.it

**Stefano Leonardi**
Sapienza University of Rome
leonardi@diag.uniroma1.it

**Rebecca Reiffenhäuser**
Sapienza University of Rome
rebeccar@diag.uniroma1.it

In this appendix, we include all the material missing from the main paper. For ease of reference, we repeat all the statements as well as the pseudocode of ADAPTIVEGREEDY.

Further, for completeness, in Section D we show that the plain density greedy is a 3-approximation algorithm in the monotone case but only a $\Theta(n)$-approximation in the non-monotone case.

## A    Missing Material from Section 3

**Proposition 1** (Nemhauser et al. [5]). *Given a function $v$, defined on $2^A$ for some set A, the following are equivalent*

(i)  $v(i\,|\,S) \geq v(i\,|\,T)$ *for all $S \subseteq T \subseteq A$ and $i \notin T$.*

(ii)  $v(S) + v(T) \geq v(S \cup T) + v(S \cap T)$ *for all $S, T \subseteq A$.*

(iii)  $v(T) \leq v(S) + \sum_{i \in T \setminus S} v(i\,|\,S) - \sum_{i \in S \setminus T} v(i\,|\,S \cup T \setminus \{i\})$ *for all $S, T \subseteq A$.*

Moreover, we restate a key result which connects random sampling and submodular maximization. The original version of the theorem was due to Feige et al. [2], although here we use a variant from Buchbinder et al. [1].

**Lemma 1** (Lemma 2.2. of Buchbinder et al. [1]). *Let $v : 2^A \rightarrow \mathbb{R}_+$ be a submodular set function, let $X \subseteq A$ and let $X(p)$ be a sampled subset, where each element of $X$ appears with probability at most $p$ (not necessarily independent). Then:*

$$\mathbb{E}\left[v(X(p))\right] \geq (1 - p)v(\emptyset).$$

As discussed in the main text, to facilitate our analysis, we state an equivalent algorithm, where the sampling phase does not proceed the greedy phase but rather it happens as the algorithm greedily considers each item. The two algorithms are equivalent in the sense that they have identical output distributions.

---
$\text{SAMPLEGREEDY}(A, v, \mathbf{c}, B)$

**1**   $i^* \in \arg\max_{k \in A} v(k)$                                                    `/* best single item */`
**2**   $S = \emptyset$                                                            `/* greedy solution */`
**3**   $F = \{k \in A | v(k) > 0\}$                          `/* initial set of feasible items */`
**4**   $R = B$                                  `/* remaining knapsack capacity */`
**5**   **while** $F \neq \emptyset$ **do**
**6**      Let $i \in \arg\max_{k \in F} \dfrac{v(k \,|\, S)}{c_k}$
**7**      Let $r_i \sim \text{Bernoulli}(p)$                              `/* independent random bit */`
**8**      **if** $r_i = 1$ **then**
**9**          $S = S \cup \{i\}$
**10**        $R = R - c_i$
**11**      $A = A \setminus \{i\}$
**12**      $F = \{k \in A \,|\, v(k \,|\, S) > 0 \text{ and } c_k \leq R\}$
**13**   **return** $\max\{v(i^*), v(S)\}$
---

## A.1   Proof of Theorem 1

**Theorem 1.** *For $p = \sqrt{2} - 1$, SAMPLEGREEDY is a $\left(3 + 2\sqrt{2}\right)$-approximation algorithm.*

*Proof.* For the analysis of the algorithm we are going to use the auxiliary set $O$, an extension of the set $S$ that respects the knapsack constraint and uses feasible items from an optimal solution. In particular, let $S^*$ be an optimal solution and let $s_1, s_2, \ldots, s_r$ be its elements in decreasing order with respect to their cost, i.e., $c_{s_1} \geq \ldots \geq c_{s_r}$. Then, $O$ is a *fuzzy* set that is initially equal to $S^*$ and during each iteration of the *while* loop it is updated as follows:

- If $r_i = 1$, then $O = O \cup \{i\}$. In case this addition violates the knapsack constraint, i.e., $c(O) > B$, then we repetitively remove items from $O \setminus S$ in increasing order with respect to their cost until the cost of $O$ becomes exactly $B$. Note that this means that the last item removed may be removed only partially. More precisely, if $c(O) > B$ and $c(O \setminus \{s_j\}) \leq B$, where $s_j$ is the item of $S^*$ of maximum index in $O \setminus S$, then we keep a $\left(B - c(O) + c_j\right)/c_j$ fraction of $s_j$ in $O$ and stop its update for the current iteration.

- Else (i.e., if $r_i = 0$), $O = O \setminus \{i\}$.

If an item $j$ was considered (in line 6) in some iteration of the *while* loop, then let $S_j$ and $O_j$ denote the sets $S$ and $O$, respectively, at the beginning of that iteration. Moreover, let $O'_j$ denote $O$ at the end of that iteration. If $j$ was never considered, then $S_j$ and $O_j$ (or $O'_j$) denote the final versions of $S$ and $O$, respectively. In fact, in what follows we exclusively use $S$ and $O$ for their final versions.

It should be noted that, for all $j \in A$, $S_j \subseteq O_j$ and also no item in $O_j \setminus S_j$ has been considered in any of the previous iterations of the *while* loop.

Before stating the next lemma, let us introduce some notation for the sake of readability. Note that, by construction, $O \setminus S$ is either empty or consists of a single fractional item $\hat{\imath}$. In case $O \setminus S = \emptyset$, by $\hat{\imath}$ we denote the last item removed from $O$. For every $j \in A$, we define $Q_j = O_j \setminus (O'_j \cup S \cup \{\hat{\imath}\})$. Note that if $j$ was never considered during the execution of the algorithm, then $Q_j = \emptyset$.

**Lemma 2.** *For every realization of the Bernoulli random variables, it holds that*

$$v(S \cup S^*) \leq v(S) + v(\hat{\imath}) + \sum_{j \in A} c(Q_j) \frac{v(j \,|\, S_j)}{c_j}.$$

*Proof of Lemma 2.* Assume that the random bits $r_1, r_2, \ldots$ are fixed. Also, without loss of generality, assume the items are numbered according to the order in which they are considered by SAMPLE-GREEDY, with the ones not considered by the algorithm numbered arbitrarily (but after the ones considered). That is, item $j$—if considered—is the item considered during the $j^{th}$ iteration.

Consider now any round $j$ of the while loop of SAMPLEGREEDY. An item is removed from $O_j$ in two cases. First, it could be item $j$ itself that was originally in $S^*$ but $r_j = 0$ (and hence it will never get back in $O_k$ for any $k > j$). Second, it could be some other item that was in $S^*$ and is taken out to make room for the new item $j$. In the latter case the only possibility for the removed item to return in $O_k$ for some $k > j$ is to be selected by the algorithm and inserted in $S$. We can hence conclude that $Q_j \cap Q_k = \emptyset$ for all $j \neq k$. In addition to that, it is clear that $S \cup S^* = S \cup \{\hat{\imath}\} \cup \bigcup_j Q_j$.

Therefore, if items $1, 2, \ldots, \ell$ where all the items ever considered, using submodularity and the fact that $S_j \subseteq S \subseteq S \cup \bigcup_{r=j+1}^{\ell} Q_r$, we have

$$v(S \cup S^*) - v(S) - v(\hat{\imath}) \leq v((S \cup S^*) \setminus \{\hat{\imath}\}) - v(S) = \sum_{j=1}^{\ell} v\left(Q_j \mid S \cup \bigcup_{r=j+1}^{\ell} Q_r\right)$$

$$\leq \sum_{j=1}^{\ell} v\left(Q_j \mid S_j\right) \leq \sum_{j=1}^{\ell} \sum_{x \in Q_j} \frac{v(x \mid S_j)}{c_x} \cdot c_x$$

$$\leq \sum_{j=1}^{\ell} \sum_{x \in Q_j} \frac{v(j \mid S_j)}{c_j} \cdot c_x = \sum_{j=1}^{\ell} \frac{v(j \mid S_j)}{c_j} \cdot \sum_{x \in Q_j} c_x$$

$$= \sum_{j=1}^{\ell} \frac{v(j \mid S_j)}{c_j} \cdot c(Q_j), \tag{1}$$

where in a slight abuse of notation we consider $c_x$ to be the fractional (linear) cost if $x \in Q_j$ is a fractional item. While the first three inequalities directly follow from the submodularity of $v$, for the last inequality we need to combine the optimality of $v(j \mid S_j)/c_j$ at the step $j$ was selected with the fact that every single item $x$ appearing in the sum $\sum_{j=1}^{\ell} \sum_{x \in Q_j} v(x \mid S_j)$ was feasible (as a whole item) at that step. The latter is true because of the way we remove items from $O$. If $x$ is removed, it is removed before (any part of) $\hat{\imath}$ is removed. Thus, $x$ is removed when the available budget is still at least $c_{\hat{\imath}}$. Given that $c_x \leq c_{\hat{\imath}}$, we get that $x$ is feasible until removed.

To conclude the proof of the Lemma it is sufficient to note that $c(Q_j) = 0$ for all items that were not considered. $\qquad \square$

While the previous Lemma holds for *each* realization of the random coin tosses in the algorithm, we next consider inequalities holding in expectation over the randomness of the $\{r_i\}_{i=1}^{|A|}$ in SAMPLE-GREEDY. The indexing of the elements is hence to be considered deterministic and fixed in advance, not as in the proof of Lemma 2.

**Lemma 3.** $\mathbb{E}\left[\sum_{j \in A} c(Q_j) \frac{v(j \mid S_j)}{c_j}\right] \leq \frac{\max\{p, 1-p\}}{p} \mathbb{E}[v(S)]$

*Proof of Lemma 3.* For all $i \in A$, we define $G_i$ to be the random gain because of $i$ at the time $i$ is added to the solution ($G_i = v(i \mid S_i)$ if $i$ is added and $0$ otherwise)

Since $v(S) = \sum_{i \in A} G_i$, by linearity, it suffices to show that the following inequalities hold in expectation over the coin tosses:

$$c(Q_i) \frac{v(i \mid S_i)}{c_i} \leq \frac{\max\{p, 1-p\}}{p} G_i, \quad \forall i \in A. \tag{2}$$

In order to achieve that, following [3], let $\mathcal{E}_i$ be any event specifying the random choices of the algorithm up to the point $i$ is considered (if $i$ is never considered, $\mathcal{E}_i$ captures all the randomness). If $\mathcal{E}_i$ is an event that implies $i$ is not considered, then the Eq. (2) is trivially true, due to $G_i = 0$ and $Q_i = \emptyset$. We focus now on the case $\mathcal{E}_i$ implies that $i$ is considered. Analyzing the algorithm, it is clear that

$$\mathbb{E}\left[c(Q_i) \mid \mathcal{E}_i\right] \leq \begin{cases} 0 \cdot \mathbb{P}\left(r_i = 1\right) + c_i \cdot \mathbb{P}\left(r_i = 0\right) = (1-p) \cdot c_i, & \text{if } i \in O_i, \\ c_i \cdot \mathbb{P}\left(r_i = 1\right) + 0 \cdot \mathbb{P}\left(r_i = 0\right) = p \cdot c_i, & \text{otherwise.} \end{cases} \tag{3}$$

In short, $\mathbb{E}\left[c(Q_i)\,|\,\mathcal{E}_i\right] \le \max\{p, 1-p\} \cdot c_i$. It is here that we use the fuzziness of $O$: without the fractional items it would be hopeless to bound $c(Q_t)$ with $c_t$.

At this point, we exploit the fact that $\mathcal{E}_i$ contains the information on $S_i$, i.e., $S_i = S_i(\mathcal{E}_i)$ deterministically. Recall that $S_i$ is the solution set at the time item $i$ is considered by the algorithm.

$$\mathbb{E}\left[G_i\,|\,\mathcal{E}_i\right] = \mathbb{P}\left(i \in S\,|\,\mathcal{E}_i\right) v(i\,|\,S_i) = \mathbb{P}\left(r_i = 1\right) v(i\,|\,S_i) = p \cdot c_i\, \frac{v(i\,|\,S_i)}{c_i}$$

$$\ge \frac{p}{\max\{p, 1-p\}}\, \mathbb{E}\left[c(Q_i)\,|\,\mathcal{E}_i\right]\, \frac{v(i\,|\,S_i)}{c_i}$$

$$= \frac{p}{\max\{p, 1-p\}}\, \mathbb{E}\left[c(Q_i)\, \frac{v(i\,|\,S_i)}{c_i}\,\bigg|\,\mathcal{E}_i\right].$$

We can hence conclude the proof by using the law of total probability over $\mathcal{E}_i$ and the monotonicity of the conditional expectation:

$$\mathbb{E}\left[G_i\right] = \mathbb{E}\left[\mathbb{E}\left[G_i\,|\,\mathcal{E}_i\right]\right] \ge \mathbb{E}\left[\frac{p}{\max\{p, 1-p\}}\, \mathbb{E}\left[c(Q_i)\, \frac{v(i\,|\,S_i)}{c_i}\,\bigg|\,\mathcal{E}_i\right]\right] =$$

$$= \frac{p}{\max\{p, 1-p\}}\, \mathbb{E}\left[c(Q_i)\, \frac{v(i\,|\,S_i)}{c_i}\right].$$

$\square$

**Lemma 4.** $v(S^*) \le \dfrac{1}{1-p}\, \mathbb{E}\left[v(S \cup S^*)\right]$.

*Proof of Lemma 4.* Let $S^*$ be an optimal set for the constrained submodular maximization problem. We define $g : 2^A \to \mathbb{R}_+$ as follows: $g(B) = v(B \cup S^*)$. It is a simple exercise to see that such function is indeed submodular, moreover $g(\emptyset) = v(S^*)$. If we now apply Lemma 1 to $g$, observing that the elements in the set $S$ output by the algorithm are chosen with probability at most $p$, we conclude that:
$$\mathbb{E}\left[v(S \cup S^*)\right] = \mathbb{E}\left[g(S)\right] \ge (1-p)g(\emptyset) = (1-p)v(S^*).$$

$\square$

Combining Lemmata 2, 3, and 4 we get

$$(1-p)v(S^*) \le \mathbb{E}\left[v(S \cup S^*)\right]$$

$$\le \mathbb{E}\left[v(S) + v(\hat{\imath}) + \sum_{j \in A} c(Q_j)\, \frac{v(j\,|\,S_j)}{c_j}\right]$$

$$\le \mathbb{E}\left[v(S)\right] + v(i^*) + \frac{\max\{p, 1-p\}}{p}\, \mathbb{E}\left[v(S)\right]$$

$$= \max\left\{2, \tfrac{1}{p}\right\} \cdot \mathbb{E}\left[v(S)\right] + v(i^*). \tag{4}$$

By substituting $\sqrt{2}-1$ for $p$, this yields $v(S^*) \le (3 + 2\sqrt{2}) \max\{\mathbb{E}\left[v(S)\right], v(i^*)\}$. This establishes the claimed approximation factor. $\square$

## A.2 Proof Sketches of Theorems 2 and 3

**Theorem 2.** *The lazy version of* SAMPLEGREEDY *achieves an approximation factor of* $3 + 2\sqrt{2} + \varepsilon$ *using* $O(n\varepsilon^{-1} \log{(n/\varepsilon)})$ *value oracle calls.*

*Proof.* For a given $\varepsilon \in (0, 1)$ let $\varepsilon' = \varepsilon/6$. We perform lazy evaluations using $\varepsilon'$. Assume that by $\log$ we denote the binary logarithm.

It is straightforward to argue about the number of value oracle calls. Since the marginal value of each element $i$ has been updated at most $\frac{\log(n/\varepsilon')}{\varepsilon'}$ times, we have a total of at most $n\frac{\log(n/\varepsilon')}{\varepsilon'} = O\left(\frac{n\log(n/\varepsilon)}{\varepsilon}\right)$ function evaluations.

The approximation ratio is also easy to show. There are two distinct sources of loss in approximation. We first bound the total value of the discarded elements due to too many updates. This value appears as the upper bound of an extra additive term in Eq. (1). Indeed, now besides $\sum_{j=1}^{\ell} v\left(Q_j \mid S \cup \bigcup_{r=j+1}^{\ell} Q_r\right)$ we need to account for the elements of $O$ that were ignored because of too many updates. Such elements, once they become "inactive" do not contribute to the cost of the current $O$ and are never pushed out as new elements come into $S$. The definition of the $Q_j$s in the proof of Theorem 1 should be adjusted accordingly. That is, if $W_j$ are the elements of $O$ that become inactive because they were updated too many times during iteration $j$, we have

$$v((S \cup S^*) \setminus \hat{\imath}) - v(S) \le \sum_{j=1}^{\ell} v\left(Q_j \mid S_j\right) + \sum_{j=1}^{\ell} v\left(W_j \mid S_j\right).$$

However, by noticing that for $x \in (0,1)$ it holds that $x \le \log(1+x)$, we have

$$\sum_{j=1}^{\ell} v\left(W_j \mid S_j\right) \le \sum_{i \in \bigcup_j W_j} (1+\varepsilon')^{-\frac{\log(n/\varepsilon')}{\varepsilon'}} v(i)$$
$$\le \sum_{i \in \bigcup_j W_j} (1+\varepsilon')^{-\frac{\log(n/\varepsilon')}{\log(1+\varepsilon')}} \max_{k \in A} v(k)$$
$$\le \sum_{i \in A} (1+\varepsilon')^{-\log_{1+\varepsilon'}(n/\varepsilon')} v(S^*)$$
$$= \sum_{i \in A} \frac{\varepsilon}{6n} v(S^*) = \frac{\varepsilon}{6} v(S^*).$$

For the second source of loss in approximation, recall that the marginals only decrease due to submodularity. So, we know that if some item $j$ is considered during iteration $j$ (following the renaming of Lemma 2), then $(1+\varepsilon')v(j \mid S_j)/c_j \ge \arg\max_{k \in F} v(k \mid S_j)/c_k$. The only difference this makes (compared to the proof of Theorem 1) is that in the last inequality of Eq. (1) we have an extra factor of $1 + \varepsilon'$.

Combining the above, we get the following analog of Lemma 2:

$$v(S \cup S^*) \le v(S) + v(\hat{\imath}) + \frac{\varepsilon}{6} v(S^*) + \sum_{j \in A} \left(1 + \frac{\varepsilon}{6}\right) c(Q_j) \frac{v(j \mid S_j)}{c_j},$$

which carries over to Eq. (4), while Lemmata 3 and 4 are not affected at all. It is then a matter of simple calculations to see that for $p = \sqrt{2}-1$, we still get $v(S^*) \le (3+2\sqrt{2}+\varepsilon)\max\{\mathbb{E}\left[v(S)\right], v(i^*)\}$. □

**Theorem 3.** *If $\max_{i \in A} v(i) \le \delta \cdot$ OPT for $\delta \in (0, 1/2)$, then* SAMPLEGREEDY *with $p = \frac{1-\delta}{2}$ is a $(4 + \varepsilon_\delta)$-approximation algorithm, where $\varepsilon_\delta = \frac{4\delta(2-\delta)}{(1-\delta)^2}$.*

*Proof.* Starting from Eq. (4) and exploiting the large instance property, we get:

$$(1-p)v(S^*) \le \max\left\{2, \frac{1}{p}\right\} \cdot \mathbb{E}\left[v(S)\right] + v(i^*) \le \max\left\{2, \frac{1}{p}\right\} \cdot \mathbb{E}\left[v(S)\right] + \delta \cdot v(S^*).$$

Rearranging the terms and assuming $p + \delta < 1$, we have:

$$v(S^*) \le \frac{\max\left\{2, \frac{1}{p}\right\}}{1 - p - \delta}.$$

Optimizing for $p \in (0, 1-\delta)$ we get the desired statement. □

# B  Missing Material from Section 4

Before proving that ADAPTIVEGREEDY works as promised, we need some observations.

Let us denote by $S$ the output of a run of our algorithm, and $S^*$ the output of a run of the optimal adaptive strategy. Fix a realization $\omega \in \Omega$. Now, Lemma 1 of [4] implies

$$\mathbb{E}\left[v(S \cup S^*, \omega) \,|\, \omega\right] \geq (1 - p) \cdot v(S^*, \omega).$$

Since $\omega$ (and therefore, $S^*$) is fixed, the only randomness is due to the coin flips in our algorithm. We stress that the union between $S$ and $S^*$ has to be intended in the following sense: run our algorithm, and independently, also the optimal one, both for the same realization $\omega$. The previous inequality is true for any $\omega$. So, by the law of total probability, we also have

$$\mathbb{E}\left[v(S \cup S^*)\right] \geq (1 - p) \cdot \mathbb{E}\left[v(S^*)\right]. \tag{5}$$

For the next observation, assume our algorithm has picked (and therefore observed) exactly set $S$. That is, we know only $\omega_{|S}$. We number all items $a \in A$ with positive marginal with respect to $S$ by decreasing ratio $v\left(a \,|\, (S, \omega_{|S})\right)/c_a$, i.e.,

$$a_1 = \arg\max_{a \in A} \left\{ \frac{v\left(a \,|\, (S, \omega_{|S})\right)}{c_a} \right\}, \quad a_2 = \arg\max_{a \in A \setminus \{a_1\}} \left\{ \frac{v\left(a \,|\, (S, \omega_{|S})\right)}{c_a} \right\}$$

and so on. Note that this captures a notion of the *best-looking items after already adding $S$*.

For $k = \min\{i \in \mathbb{N} \,|\, \sum_{l=1}^{i} c_l \geq B\}$, we get, in analogy to Lemma 1 of Gotovos et al. [4],

$$\sum_{i=1}^{k} v\left(a_i \,|\, (S, \omega_{|S})\right) \geq \mathbb{E}\left[ \sum_{a \in S^*} v\left(a \,|\, (S, \omega_{|S})\right) \,\Big|\, \omega_{|S} \right] \geq v\left(S^* \,|\, (S, \omega_{|S})\right). \tag{6}$$

For the sake of the future analysis the last element $k$ is considered fractionally, so that $\sum_{i=1}^{k} c_i = B$. Note that it could be the case that $k$ is not well defined, as there may not be enough elements with positive marginal to fill the knapsack. If that is the case, just consider $k$ to be the number of elements with positive marginals.

The point of (6) is that, given $(S, \omega_{|S})$, the set of elements $a_1 \ldots a_k$ is deterministic, while $S^*$ is not, because it corresponds to the set opened by the best adaptive policy. Moreover, in the middle term notice that the conditioning influences the valuation, but *not* the policy, since we are assuming to run it obliviously. This is fundamental for the analysis.

Since this holds for any set $S$, we can again generalize to the expectation over all possible runs of the algorithm (fixing the coin flips or not, as they only influence $S$; the best adaptive policy or the best-looking items $a_1, a_2, \ldots, a_k$ are not affected). So, we get

$$\mathbb{E}\left[ \sum_{i=1}^{k} v\left(a_i \,|\, (S, \omega_{|S})\right) \right] \geq \mathbb{E}\left[ v\left(S^* \,|\, (S, \omega_{|S})\right) \right]. \tag{7}$$

We remark that $k$ above is a random variable which depends on $S$. We use these observations to prove the ratio of our algorithm.

## B.1  Proof of Theorem 4

**Theorem 4.** *For $p_0 = 1/3$ and $p = 1/6$, ADAPTIVEGREEDY yields a 9-approximation of $\mathrm{OPT}_\Omega$, while its lazy version achieves a $(9 + \varepsilon)$-approximation using $O(n\varepsilon^{-1} \log (n/\varepsilon))$ value oracle calls. Moreover, when $\max_{i \in A} v(i) \leq \delta \cdot \mathrm{OPT}_\Omega$ for $\delta \in (0, 1/2)$, then for $p_0 = 0$ and $p = (\sqrt{3 - 2\delta} - 1)/2$, ADAPTIVEGREEDY yields a $(4 + 2\sqrt{3} + \varepsilon'_\delta)$-approximation, where $\varepsilon'_\delta \approx \frac{6\delta(2-\delta)}{(1-\delta)^2}$.*

*Proof.* For any run of the algorithm, i.e., a fixed set $S$, the corresponding partial realization $\omega_{|S}$ and the coin flips observed, define for convenience the set $C$ as those items in $\{a_1, \ldots, a_k\}$ that have been considered during the algorithm and then not added to $S$ because of the coin flips. Define $U = \{a_1, \ldots, a_k\} \setminus C$. Additionally, define $C'$ to be the set of all items that are considered, but not

---

ADAPTIVEGREEDY

---

1  Let $r_0 \sim \mathrm{Bernoulli}(p_0)$

2  **if** $r_0 = 1$ **then**

3     |  $i^* \in \arg\max_{k \in A} v(k)$                 /* `best single item in expectation` */

4     |  Observe $\omega_{i^*}$ and **return** $v(i^*, \omega_{i^*})$

5  $S = \emptyset, R = B$               /* `greedy solution and remaining knapsack capacity` */

6  $F = \{k \in A \mid v(k) > 0\}$            /* `initial set of candidate items` */

7  **while** $F \neq \emptyset$ **do**

8     |  Let $i \in \arg\max_{k \in F} \frac{v(k \mid (S, \omega_{|S}))}{c_k}$

9     |  Let $r_i \sim \mathrm{Bernoulli}(p)$             /* `independent random bit` */

10     |  **if** $r_i = 1$ **then**

11     |    |  Observe $\omega_i :$  $S = S \cup \{i\}, \ R = R - c_i$

12     |  $A = A \setminus \{i\}, \quad F = \big\{ k \in A \mid v(k \mid (S, \omega_{|S})) > 0 \text{ and } c_k \leq R \big\}$

13  **return** $S, v(S, \omega_{|S})$

---

chosen during the run of our algorithm which have positive expected marginal contribution to $S$. I.e., $C$ captures the items from the *good-looking* set after choosing $S$ that we missed due to coin tosses, and $C'$ *all* items we missed for the same reason which should have had a positive contribution in hindsight. Note that $C \subseteq C'$

We can then split the left hand side term of (7) into two parts: the sum over $C$ (upper bounded by the sum over $C'$), and the sum over $U$. Now we control separately these terms using linear combinations of $v(S)$ and $v(i^*)$.

**Lemma 5.** $\mathbb{E}[v(S)] \geq p \cdot \mathbb{E}\left[\sum_{a \in C} v\big(a \mid (S, \omega_{|S})\big)\right]$

*Proof.* Since $C \subseteq C'$ and $C'$ contains all considered elements with nonnegative expected contribution to $S$, it is sufficient to show $\mathbb{E}[v(S)] \geq p \cdot \mathbb{E}\left[\sum_{a \in C'} v\big(a \mid (S, \omega_{|S})\big)\right].$

We proceed as in Lemma 3. Let's consider for each $a \in A$ all the events $\mathcal{E}_a$ capturing the story of a run of the algorithm up to the point element $a$ is considered (all the history if it is never considered).

Let $G_a$ be the marginal contribution of element $a$ to the solution set $S$. If $\mathcal{E}_a$ corresponds to a story in which element $a$ is not considered, then it does not contribute - neither in the left, nor in the right hand side of the inequality we are trying to prove. Else, let $(S_a, \omega_a)$ be the partial solution when it is indeed considered:

$$\mathbb{E}[G_a \mid \mathcal{E}_a] = p \cdot v(a \mid (S_a, \omega_a)) \geq p \cdot \mathbb{E}\big[v(a \mid (S, \omega_{|S})) \mid \mathcal{E}_a\big].$$

The statement follows from the law of total probability with respect to $\mathcal{E}_a$, and state-wise submodularity of $v$. $\qquad\square$

**Lemma 6.** $\mathbb{E}[v(S)] + v(i^*) \geq \mathbb{E}\left[\sum_{a \in U} v\big(a \mid (S, \omega_{|S})\big)\right].$

*Proof.* Now let us turn towards the items $U$ that were not considered by the algorithm. The intuition behind the claim is that if they were not considered then they were not good enough, in expectation, to compare with $S$. The proof, though, has to deal with some probabilistic subtleties.

Let's start fixing a story of the algorithm, i.e., the coin tosses and $(S, \omega_S)$, $S = s_1, s_2, \ldots, s_T$, numbered according to their insertion in S, i.e., $s_i$ is the $i^{th}$ element to be added to $S$. For the sake of simplicity let's also renumber the elements in $U$ as $a_1, \ldots, a_l$ respecting the order given by the marginals over costs.

There are two cases. If during the whole algorithm the elements in $U$ have ratio $\frac{v\big(a \mid (S_i, \omega_{|S_i})\big)}{c_a}$ smaller than that of the item which was instead considered, then one can easily argue, by adaptive submodularity, that:

$$\sum_{a \in U} v\left(a \mid (S, \omega_{\mid S})\right) \leq \sum_{t=1}^{T} v(s_t \mid (S_t, \omega_t)) + v(u_1 \mid (S, \omega_{\mid S})) \leq$$

$$\leq \sum_{t=1}^{T} v(s_t \mid (S_t, \omega_t)) + v(u_1) \leq \sum_{t=1}^{T} v(s_t \mid (S_t, \omega_t)) + v(i^*).$$

being $S_t = (s_1, \ldots, s_{t-1})$ and $\omega_t$ the restriction of $\omega_{\mid S}$ to $S_t$. Note that the last element $u_1$ is added to account for the unspent budget by the solution: the first inequality holds because our solution fills all the budget (up to *at most* one item) with densities which are better than all the $v\left(a \mid (S, \omega_{\mid S})\right)$ We claim that the above inequality holds also in the case in which there is an element in $U$ whose marginal over cost is greater than that of some in $S$. Such an element can exist because of the budget constraint: during the algorithm it had better marginal over cost, but was discarded because there was not enough room for it. We observe there can exist at most one such element, due to the budget constraint and because its value is upper bounded by $u_1$, so the above formula still holds.

Once we know that, by law of total probability, we have

$$\mathbb{E}\left[\sum_{a \in U} v\left(a \mid (S, \omega_{\mid S})\right)\right] \leq \mathbb{E}\left[v(S)\right] + v(i^*), \tag{8}$$

concluding the proof. $\qquad\square$

Combining the two Lemmata we get:

$$\left(1 + \tfrac{1}{p}\right) \mathbb{E}\left[v(S)\right] + \mathbb{E}\left[v(i^*)\right] \geq \mathbb{E}\left[\sum_{a \in U} v\left(a \mid (S, \omega_{\mid S})\right)\right] + \mathbb{E}\left[\sum_{a \in C} v\left(a \mid (S, \omega_{\mid S})\right)\right] =$$

$$= \mathbb{E}\left[\sum_{a \in U \cup C} v\left(a \mid (S, \omega_{\mid S})\right)\right].$$

Equation (7) implies

$$\left(1 + \tfrac{1}{p}\right) \mathbb{E}\left[v(S)\right] + \mathbb{E}\left[v(i^*)\right] \geq \mathbb{E}\left[\sum_{a \in U \cup C} v\left(a \mid (S, \omega_{\mid S})\right)\right] \geq \mathbb{E}\left[v\left(S^* \mid (S, \omega_{\mid S})\right)\right].$$

Also, with some rewriting and Equation (5):

$$\mathbb{E}\left[v\left(S^* \mid (S, \omega_{\mid S})\right)\right] = \mathbb{E}\left[\mathbb{E}\left[v(S^* \cup S, \omega) - v(S, \omega) \mid \omega_{\mid S}\right]\right]$$
$$= \mathbb{E}\left[v(S^* \cup S)\right] - \mathbb{E}\left[v(S)\right]$$
$$\geq (1 - p) \cdot \mathbb{E}[v(S^*)] - \mathbb{E}\left[v(S)\right]$$

All together, denoting as OPT the $\mathbb{E}[v(S^*)]$, we get:

$$(2p + 1)\mathbb{E}[v(S)] + p\mathbb{E}[v(i^*)] \geq p(1 - p) \, \text{OPT} \tag{9}$$

Let's call ALG the expected value of the solution output by the algorithm. Since the algorithm chooses with a coin flip either the best expected single item or $S$, it holds

$$\text{ALG} = (1 - p_0)v(S) + p_0 v(i^*)$$

Picking $p_0 = \frac{p}{3p+1}$,

$$\text{ALG} = \frac{2p + 1}{3p + 1} \mathbb{E}[v(S)] + \frac{p}{3p + 1} \mathbb{E}[v(i^*)] \geq \frac{p(1 - p)}{3p + 1} \, \text{OPT} \, .$$

The right hand side is minimized for $p = \frac{1}{3}$, concluding the proof of the first part of the statement.

The lazy version of ADAPTIVEGREEDY is analogous to the non-adaptive setting, both for the algorithm and the analysis, so we omit repeating the proof.

In order to prove the last part of the statement, we start from Eq. (9) and apply the large instance property:

$$(1-p)\mathbb{E}\left[v(S^*)\right] \leq \left(2 + \tfrac{1}{p}\right)\mathbb{E}\left[v(S)\right] + \mathbb{E}\left[v(i^*)\right] \leq \left(2 + \tfrac{1}{p}\right)\mathbb{E}\left[v(S)\right] + \delta \cdot \mathbb{E}\left[v(S^*)\right]$$

Rearranging terms and assuming $p + \delta < 1$ we have that:

$$\mathbb{E}\left[v(S^*)\right] \leq \frac{\left(2 + \tfrac{1}{p}\right)}{1 - p - \delta} \cdot \mathbb{E}\left[v(S)\right].$$

Optimizing for $p \in (0, 1 - \delta)$, we get the claimed result. Specifically, for $p = (\sqrt{3 - 2\delta} - 1)/2$ the approximation factor is $(4 + 2\sqrt{3} + \varepsilon_\delta)$, with

$$\varepsilon_\delta = 2\left(\frac{\sqrt{3 - 2\delta} + 1}{(1 - \delta)^2} + \frac{1}{1 - \delta} - \sqrt{3} - 2\right) \approx \frac{6\delta(2 - \delta)}{(1 - \delta)^2}.$$

$\square$

## C  Additional Details on Section 5

Before moving on to providing additional information for each specific experiment, we note that the computational setup consisted of a 13-inch 2014 MacBook Pro with a 2,6 GHz Intel Core i5 processor and 8GB of RAM. Moreover, all graphs contain error bars, indicating the standard deviation between different runs of the experiments. This is usually insignificant due to the concentrating effect of the large size of the instances, despite the randomly initialized weights and inherent randomness of the algorithms used. Nevertheless, all results are obtained by running each experiment a number of times. For all algorithms involved, we use *lazy evaluations* with $\varepsilon = 0.01$.

**Video Recommendation:**  We expand on the exact definition of the similarity measure that is only tersely described in the main text. Each movie $i$ is associated with a tag vector $t^i \in [0, 1]^{1128}$, where each coordinate contains a relevance score for that individual tag. These tag vectors are *not* normalized and have no additional structure, other than each coordinate being restricted to $[0, 1]$. We define the similarity $w_{ij}$ between two movies $i, j$ as:

$$w_{ij} = \sqrt{\sum_{k=1}^{1128}\left(\min\{t^i_k, t^j_k\}\right)^2}.$$

In other words, it is the L2 norm of the *coordinate-wise minimum* of $t^i$ and $t^j$. This metric was chosen so that if *both* movies have a high value in some tag, this counts as a much stronger similarity than one having a high value and the other a low one. For example, if we consider an inner product metric, any movie with all tags set to 1 would be as similar as possible to all other movies, even though it would include many tags that would be missing from the others. In particular, any movie would be appear more similar to the all 1 movie than to itself! Choosing the minimum of both tags avoids this issue. Another possibility would be to normalize each tag vector before taking the inner product, to obtain the *cosine similarity*. Although this alleviates some of the issues, there is some information loss as one movie could meaningfully have higher scores in all tags than another one; tags are not mutually exclusive. Ultimately any sensible metric has advantages and disadvantages and the exact choice has little bearing on our results. The similarity scores are then divided by their maximum as a final normalization step.

The experiment was repeated 5 times. The budget is represented as a fraction of the total cost starting at 1/100 and geometrically increasing to 1/10 in 10 steps. The total computation time was around 3 hours.

**Influence-and-Exploit Marketing:** For the YouTube graph, the experiment was repeated 5 times for a budget starting at 1/100 of the total cost and geometrically increasing to 1/3 in 20 steps, leading to a total computation time of 7 hours. For the Erdős–Rényi graph with $n$ vertices and edge probability $5/\sqrt{n}$ it was repeated 10 times, for $n$ starting at 50 and geometrically increasing to 2500 in 20 steps, taking approximately 10 minutes.

**Maximum Weighted Cut:** The experiment was repeated 10 times for $n$ starting at 10 and increasing geometrically to 300 in 20 steps, requiring approximately 5 minutes.

## D  Approximation Ratio of Deterministic Density Greedy

Here we show the 3-approximation given by the deterministic density greedy to the monotone problem and we provide a counterexample illustrating its poor performance in the non-monotone case. The pseudocode is given below; the blue parts are only for the analysis. By $S^*$ we denote an optimal solution.

---

**Algorithm 3:** DENSITYGREEDY

---

1 **Input:** set $\mathcal{A}$, $v$ monotone submodular function on $2^A$, budget $B$
2 $S \leftarrow \emptyset$, $X \leftarrow A$
3 $O \leftarrow S^*$ : fuzzy set; elements sorted by costs in decreasing order; initially equal to optimal solution
4 **while** $X \neq \emptyset$ **do**
5 $\quad$ let $i \in \arg\max_{x \in X} \frac{v_S(x)}{c_x}$
6 $\quad$ **if** $c(S) + c(i) \geq B$ *and* $i \in S^*$ **then**
7 $\quad\quad$ $\lfloor$ **return** $\arg\max\{v(S), \max_{i \in A} v(\{i\})\}$
8 $\quad$ **if** $c(S) + c(i) \leq B$ **then**
9 $\quad\quad$ $S \leftarrow S \cup \{i\}$
10 $\quad\quad$ Add $i$ to $O$ (if $i \notin O$ already). Make room for it, if needed, by erasing elements in $S^* \setminus S$ at the end of $O$
11 $\quad$ $\lfloor$ $X \leftarrow X \setminus \{i\}$
12 **return** $\arg\max\{v(S), \max_{i \in A} v(\{i\})\}$

---

**Theorem 5.** DENSITYGREEDY *is a* 3-*approximation algorithm when the submodular objective is monotone, but only a* $\Theta(n)$-*approximation algorithm when it is non-monotone.*

*Proof.* First assume $v$ is monotone. Let $j \in S$, then we call $S_j$ and $O_j$ the sets at the beginning of the while loop when $j$ is considered, and $O'_j$ at the end of the same iteration. Moreover we call $\hat{\imath}$ the element triggering the blue 'return'.

Some properties:

1. $S_j \subseteq O_j \quad \forall j$

2. $O \setminus S$ consists of (a possibly fractional part of) $\hat{\imath}$ because the elements in $O$ are sorted in decreasing cost, if there is no more room for one, then it is the last.

3. For each item $j \in S$, we call $Q_j$ the fractional set that was erased from $O$ to make room for it, without the items that later were added to the solution and $\hat{\imath}$: $Q_j = O_j \setminus (O'_j \cup S \cup \{\hat{\imath}\})$. Clearly $c(Q_j) \leq c_j$.

4. $Q_j \cap Q_i = \emptyset$

5. $S \cup S^* = S \cup \{\hat{\imath}\} \cup \bigcup_{j \in S} Q_j$ and the union is disjoint.

Let ALG $= \max\{v(S), \max_{i \in A} v(\{i\})\}$. Starting from the last point we have:

$$v(S^*) \leq v(S \cup S^*) \leq v(S) + v(\hat{\imath}) + \sum_{j \in S} v(Q_j \mid S) \leq v(S) + v(\hat{\imath}) + \sum_{j \in S} v(Q_j \mid S_j)$$

$$\leq v(S) + v(\hat{\imath}) + \sum_{j \in S} \sum_{x \in Q_j} \frac{v(x \mid S_j)}{c_x} c_x \leq v(S) + v(\hat{\imath}) + \sum_{j \in S} \frac{v(j \mid S_j)}{c_j} c(Q_j)$$

$$\leq v(S) + v(\hat{\imath}) + \sum_{j \in S} v(j \mid S_j) \leq 2v(S) + v(\hat{\imath})$$

$$\leq 3 \cdot \text{ALG} \,,$$

completing the proof for the monotone case.

For the non-monotone case, it is straightforward to see that DENSITYGREEDY is at least a $n$-approximation algorithm. Indeed, if $S^* = \{i_1, \dots, i_r\}$, then

$$v(S^*) = \sum_{j=1}^{r} v(i_j \mid \{i_1, \dots, i_{j-1}\}) \leq \sum_{j=1}^{r} v(i_j) \leq n \cdot \max_{i \in A} v(i) \leq n \cdot \text{ALG} \,,$$

where the first inequality follows from submodularity.

Next we define a family of non-monotone instances such that the approximation guarantee of DENSITYGREEDY cannot be better than $n - 1$. For any value of $n \in \mathbb{N}$, the $n$th instance will have $n$ elements, i.e., $A_n = \{1, 2, \dots, n\}$, all with cost 1, and available budget $B_n = n$. The objective function $v_n$ is defined as follows.

$$v(S) = \left\{ \begin{array}{ll} |S| \,, & \text{if } S \subseteq \{1, 2, \dots, n-1\} \\ 1 + n^{-1} \,, & \text{otherwise} \end{array} \right.$$

It is straightforward to check that $v_n$ is a normalized, non-negative submodular function. Moreover, it is easy to see that for its greedy solution DENSITYGREEDY would start by adding element $n$ and then it would stop. Given that $n = \arg\max_{i \in A_n} v_n(i)$, DENSITYGREEDY returns $\{n\}$ for a total value of $1 + n^{-1}$ instead of the optimal $n - 1$ (achieved by $S^* = \{1, 2, \dots, n-1\}$). So, for any fixed $\varepsilon > 0$, DENSITYGREEDY cannot guarantee at least a $(n - 1 - \varepsilon)^{-1}$ fraction of the optimal value. □