[Reviews · NeurIPS 2020]

Review 1

Summary and Contributions: In this paper, the authors study the problem of maximizing a submodular function subject to a knapsack constraint. They present a simple randomized greedy algorithm that achieves a 5.83 approximation. They also study the stochastic version of this problem. In experimental evaluations, they evaluate the performance of their algorithms on real and synthetic datasets.

Strengths: 1. It is an important problem. 2. The paper is well-written. 3. Experimental evaluations are shows the effectiveness of the proposed algorithms. The code is provided and the results are reproducible. === After author response === I have read the author's feedback and the other reviews. Given the marginal contribution of this work, I would keep my score unchanged.

Weaknesses: a. The main reason for my score is the level of the contribution of this paper. The contribution of this paper is marginal due to two factors: 1. The algorithm for the monotone function is extensively is studied although it is not the state of the art. 2. The ides of sub-sampling to convert an algorithm that is designed for monotone functions to non-monotone function is also well investigated: [3], [FKK], and [AEFNS]. Given these previous works, combining these two steps seems straightforward. Furthermore, the extension to the adaptive case is somewhat straightforward given the result of [25]. b. The authors do not use the sate of the art problem for maximizing a monotone submodular function subject to a knapsack constraint. [YZA] provides a tighter result. I think merging the idea of sub-sampling with the result of [YZA] improves the approximation guarantee. c. The idea of reducing the computational complexity by lazy evaluations is a direct consequence of the result of [36]. [YZA] Grigory Yaroslavtsev, Samson Zhou, and Dmitrii Avdiukhin. "“Bring Your Own Greedy”+ Max: Near-Optimal 1/2-Approximations for Submodular Knapsack." In International Conference on Artificial Intelligence and Statistics, pp. 3263-3274. 2020. [FKK] Moran Feldman, Amin Karbasi, and Ehsan Kazemi. "Do less, get more: Streaming submodular maximization with subsampling." In Advances in Neural Information Processing Systems, pp. 732-742. 2018. [AEFNS] Naor Alaluf, Alina Ene, Moran Feldman, Huy L. Nguyen, and Andrew Suh. "Optimal streaming algorithms for submodular maximization with cardinality constraints." In 47th International Colloquium on Automata, Languages, and Programming (ICALP 2020). Schloss Dagstuhl-Leibniz-Zentrum für Informatik, 2020

Correctness: I checked the proof for the non-adaptive case and the results are correct. As far as I checked, for the adaptive case, the proofs are correct. The empirical methodology supports the theoretical results.

Clarity: The paper is well written and easy to follow.

Relation to Prior Work: The relation to the previous work is clear. It is not discussed and is not clear to me why this result is not the direct consequence of the previous works. The reference to the state of the art algorithm for monotone submodular maximization subject to a knapsack constraint is missing. I believe that result could improve the guarantee provided by this paper too.

Reproducibility: Yes

Additional Feedback: 1. Please explain the difficulty of proving the guarantee, given the existing works on using the idea of sub-sampling for non-monotone submodular maximization. 2. Use the algorithm of [YZA] for the main building block of your algorithm.


Review 2

Summary and Contributions: The paper considers the problem of maximizing a (not necessarily monotone) submodular function subject to a knapsack constraint in the offline and adaptive settings. It provides 5.83 and 9 approximations for these two settings, respectively, using simple and efficient algorithms.

Strengths: The problem studied by the paper is natural and important, but for some reason, did not receive much attention from the research community. All the papers that refer to this problem do so only as an afterthought, while focusing mainly on a different setting. Thus, it is nice to see a paper that puts this problem in the center of the stage, and therefore, is likely to encourage more work on this somewhat forgotten problem. It should also be mentioned that the empirical results of the paper present the algorithms suggested by it in a good light, but naturally can compare these algorithms only with the state-of-the-art. Since the program has been partially forgotten, this state-of-the-art is not too impressive, and thus, being better than it is, unfortunately, not very impressive.

Weaknesses: The paper is quite incremental with respect to the techniques. In other words, it basically presents the results one can almost immediately get by throwing standard techniques of submodular maximization at the problem considered. Hence, the value of the paper is mostly in promoting the study of its problem, and providing baseline results for it on which one can improve with future research. In addition, the paper seems to make one or two assumptions that are a bit hidden, and significantly weaken the result it presents for the adaptive setting (see the detailed comments section).

Correctness: Naturally, most of the proofs have been deferred to the additional material, which I only skimmed over. However, based on the explanations given in the paper itself, it looks like everything should work out.

Clarity: Yes. Reading the paper is quite enjoyable.

Relation to Prior Work: Yes, the the relation with the previous work is discussed to a sufficient degree.

Reproducibility: Yes

Additional Feedback: - The response was read. The main question is about the novelty of the techniques used by the paper. Following the response, I took a look at the supplement material to see if there is indeed anything surprising there. However, I unfortunately failed to find such a thing, and thus, I keep my score. - On Page 2, a 2-approximation for the monotone version of the problem is attributed to [45]. I did not find in [45] such a claim, and moreover, the algorithm described does not have such a good approximation ratio (it provides about 2.58-approximation). - Section 2 is the first place mentioning the assumption in the adaptive setting that the objective function is a distribution over submodular functions. This assumption should appear already in the statement of the results. Furthermore, it is unclear from the description in Section 2 whether the paper also assumes that the objective function depends only on the state of the elements in the evaluated set. This should be clarified. Furthermore, if that is the case, then this assumption should also appear already in the statement of the results. - The experiments section assumes that the algorithm of the paper are run 5 times. The quality of the output in this mode of execution is interesting, but it would also be interesting to see the expected value of a single run and the standard deviation. Furthermore, this mode of execution does not seem to make sense in the adaptive setting. - According to the graphs in Section 5, FANTOM exhibit a super-linear dependence of the query complexity on the input size, while the algorithms of the current paper do not exhibit such a behavior. However, this seems to contradict the theoretical running time analysis presented in the paper, and thus, some discussion of this apparent contradiction will be in place. - The results of Greedy are omitted for some reason in Figure 1(b). This should be either fixed, or explained. - The conclusion expresses hope that the sampling technique will be used in the streaming settings in a future work. However, this was already done by a paper named “Do Less, Get More: Streaming Submodular Maximization with Subsampling” (NeurIPS 2018).


Review 3

Summary and Contributions: This paper is about maximizing a non-monotone submodular function subject to knapsack constraint. First of all, the paper proposes a randomized greedy algorithm which achieves a 5.83-approximation ratio and runs in O(nlogn) time (number of oracle evaluation). Although the approximation ratio is not state-of-the-art which is e=2.72, the running time is much faster and can be used in the practice. Secondly, it transfers the algorithm into a stochastic version in order to solve the adaptive submodular maximization, and obtains 9-approximation ratio. Furthermore, the paper gives the experimental comparison results in several instances like Video Recommendation, influence-and-Exploit Marketing, and Maximum Weighted Cut to show that the given algorithm is indeed practical and performs well. ==ADDED== In the response, the comparison about the ENV paper is clean. It is not practical compared to the result in this paper.

Strengths: 1. Provide a practical algorithm for non-monotone submodular maximization subject to knapsack constraint which has constant approximation ratio. The algorithm is quite simple and easy to implement. 2. Provide experimental results for their algorithm which proves that their algorithm can run well in practice. They also compare it with Fantom’s algorithm.

Weaknesses: 1. The approximation ratio is not good. 2. “To the best of our knowledge, nothing nontrivial is known for non-monotone functions and a knapsack constraint.”: There is a paper “Submodular Maximization with Matroid and Packing Constraints in Parallel” in STOC 19 which is not mentioned in this paper. In that paper, they provide a 1/e-ratio approximation algorithm for non-monotone submodular maximization under packing constraint while knapsack constraint is a special case of packing constraint. I believe the STOC paper has worse query complexity than this paper since it has heavy dependency over epsilon. But the STOC paper has very small (roughly O(log^2 n)) adaptive complexity (number of sequential rounds of independent value oracle calls). This means it is possible that this algorithm is practical in the sense that it may be feasible to run in large distributed system. Since that result considers more general constraint and matches the state-of-the-art approximation ratio which is much better than the ratio in this paper, it is worth to compare the results with that result, both in query complexity and in adaptive complexity.

Correctness: In my understanding, the theoretical results are correct.

Clarity: Overall, the structure of the paper is clear, and allows readers to clearly understand the author's intentions. One point that may cause misunderstanding is the definition of adaptive submodular in the title. The adaptive submodular here is based on the stochastic of element acquisition, not the adaptive complexity (number of sequential rounds of independent value oracle calls) in the literature. Some comments: 1. Line 51: it should be “density greedy algorithm”. 2. Line 251: “n=5 iterations”, but in the other part in the paper, n is used to represent the number of items.

Relation to Prior Work: The paper should compare their results with “Submodular Maximization with Matroid and Packing Constraints in Parallel” in STOC 19.

Reproducibility: Yes

Additional Feedback:


Review 4

Summary and Contributions: This paper considers non-monotone adaptive and non-adaptive submodular maximization under a knapsack constraint. The proposed algorithms achieve (3+2*sqrt(2))-approximation and 9-approximation for the non-adaptive and adaptive setting, respectively.

Strengths: The largest contribution of this paper is that it proposes a simple approach to non-monotone submodular maximization under a knapsack constraint. The current best approximation ratio 1/e was obtained by using the continuous greedy algorithm, which is not so practical. This paper proposes a practical, easy-to-implement greedy algorithm based on the sampling greedy algorithm by Feldman, Harshaw, and Karbasi (2018), whose approximation ratio (3+2*sqrt(2)) \approx 5.8 is best among the existing combinatorial algorithms. Moreover, the authors apply a similar approach to the setting of adaptive submodular maximization with a slightly worse approximation guarantee, which is the first algorithm for this problem. This paper seems to be a useful contribution to submodular maximization research.

Weaknesses: In my opinion, a weakness of this paper is its theoretical novelty. The analyses for the non-adaptive setting are a good combination of Feldman, Harshaw, and Karbasi (2018) and the standard technique for knapsack problems that takes the maximum of the largest singleton and the greedy solution. They are cleverly modified so that the adaptive setting can be dealt with, but there is an existing study (Gotovos, Karbasi, and Krause (2015)) on non-monotone adaptive submodular maximization that uses a similar approach as the authors mentioned. Both proof techniques are neat but not surprisingly novel. Another weakness is its practical side. To make the approximation guarantees hold, the sampling probability p should be set to the value specified in the theorems such as 0.41 or 1/6, but in the experiments, the authors set the sampling probability p in [0.9, 1], which make the proposed algorithm quite similar to the naive greedy algorithm. The authors claim that the proposed algorithms empirically perform well, but it is difficult to say the theoretical results are supported by the experimental results due to the artificial value of p.

Correctness: The technical content of the paper appears to be correct.

Clarity: The paper is generally well-organized and clearly written.

Relation to Prior Work: This paper mentions sufficiently many existing studies.

Reproducibility: Yes

Additional Feedback: In line 49 of the appendix, S \cup_{r=j*1}^\ell Q_r might be replaced with S \cup \bigcup_{r=j*1}^\ell Q_r. In line 180 of the appendix, a right parenthesis is missed. # Update after the author response I read the author response. The authors responded to each of my two concerns. (1) On technical novelty. Though applying the sub-sampling technique to the non-monotone submodular maximization under a knapsack constraint is a nice contribution, I don't think this paper develops a technically new idea. (2) On practical performance. The authors claim that they can make an artificial example in which small p works, but small p does not work well for practical objective functions. However, to show the practical advantage of the proposed algorithm, the authors should find a practical example which p smaller than 1 works. Due to these reasons, I keep my score.

[Author Response · NeurIPS 2020]

We thank the reviewers for their time and effort. Below, we address the issues raised and clarify some misconceptions.
We begin with two key issues mentioned by more than one reviewer in order to avoid repetition.

*Technical contribution.* It is indeed true that the broad existing literature on dealing with non-monotonicity by
introducing randomness creates the impression that subsampling works almost like a black-box reduction. Starting
with [6], these ideas have been used extensively for several constraints and in every possible variant of the problem.
Yet, **SAMPLEGREEDY is the first random greedy result for a knapsack constraint**. During a seven-year period,
there is not a single paper that manages to apply subsampling also for a knapsack constraint, despite the latter being as
prevalent in the submodular literature as cardinality or matroid constraints. This fact illustrates perfectly the underlying
technical difficulties: while the algorithmic idea itself is immediate, obtaining our results is not. As we mention in
Section 1.1 (lines 88-100), the tools needed to show our results differ substantially from other works, which *all* rely
on the algorithm making good progress in a step-wise fashion. This being impossible for knapsack constraints, our
main technical contribution is the *global* analysis via a comparison with a fuzzy set of high value. So, while employing
known *principles*, our work shares only few technical steps with other related work. This is also apparent when dealing
with the adaptive setting's subtleties. E.g., monotonicity is crucial in [24] and this cannot be addressed by a naive
application of subsampling. Indeed, the whole point of [25] is to make the 'subsampling lemma' work for the case of a
uniform matroid. While we share the first step of our analysis with [25], their (somewhat more complex) sampling
procedure and its analysis cannot be extended to the knapsack case in any sensible way.

*Quality of approximation.* Our clear priority is keeping the running time almost linear *and* practical; this is challenging
even for the monotone case. The fact is that **SAMPLEGREEDY is the first constant factor approximation algorithm**
**asking $O(n \log n)$ queries** for the non-monotone version of the problem. For this reason, using more elaborate
building blocks requiring $\Omega(n^2)$ queries in order to improve the approximation, was not really of interest here.
(Actually, by enumerating, like in [43], our approximation factor improves to $\frac{2e}{e-1} \approx 3.16$ with $O(n^3 \log n)$ queries.)

**Review 1.** a) Please see our remarks above. b) Thank you for bringing [YZA] to our attention. [FKK,AEFNS] are only
implicitly related (streaming setting and no knapsack constraints); nevertheless we will add all three to our related work.
c) Although this could be interesting for future work, it is not immediately clear why the analysis of [YZA] would carry
on without too much loss and yield a significantly better factor if we use their algorithm instead. Even if that was the
case, it is *impossible* to translate this to the adaptive setting, where we commit to all past choices (see also lines 33-35,
97-100). d) We do mention the state-of-the-art for the monotone case [43,16]. We assume the reviewer refers to the
streaming setting, which we do not discuss in detail as it is not directly relevant.

**Review 2.** a) Please see our remarks above. b) The algorithm in lines 52-55 is a minor variant of the $Z^H$ solution of
[45]. However, the reviewer is right about the approximation ratio: the 2 should be 2.8 as follows from [45]. c) The
value of the adaptive objective depends only on the state of the elements in the evaluated set (lines 164-165). We can
discuss this earlier on as well. Note, however, that we work on the *existing* framework of [24,25] without simplifying it.
d) Once a value of $p$ is selected, the variance is very small for large instances. Running the experiment 5 times (which
is still very fast) produces slightly better results primarily by 'guessing' a good value of $p$ (see also lines 53-54 herein).
So, even in the adaptive case, this could be seen as a dry run to tune $p$ before trying on actual data. For completeness,
we can add the expectation/variance results for a single run. e) In practice, lazy evaluations often result in much less
than $\log n$ evaluations per element. So, indeed, what we see is much closer to $n$ vs $n \log n$. f) We will add the results
for GREEDY. They were omitted to highlight that even after 5 runs, SAMPLEGREEDY has a much better performance
per number of queries ratio to FANTOM. The performance of GREEDY sits between the two. g) In our conclusions we
refer to improving streaming algorithms for knapsack constraints, e.g., [39], whereas [FKK] deals with $p$-matchoids.

**Review 3.** a) Please see our remarks above. b) We thank the reviewer for bringing the Ene et al. paper [ENV] to our
attention. It should be noted, though, that papers in this line of work do not necessarily end up having practical query
complexity. Here [ENV] assumes access to oracles for the multilinear extension of the objective and its gradient. This is
not realistic in practice (where a polytime oracle for the objective is often straightforward) but is generally dismissed as
it can be resolved via sampling. When it is done explicitly though, it requires a polynomial number of samples for each
evaluation; typically $\Omega(n^3)$ (e.g., in the seminal work of Vondrák 'Optimal approximation for the submodular welfare
problem in the value oracle model' $n^5$ samples are used per evaluation). In the case of [ENV] this results in at least
$n^4 \log^2 n$ oracle queries in the worst case. c) While we agree, there is not much flexibility about the name; 'adaptive
submodular maximization' was established even before 'adaptive complexity of submodular maximization'.

**Review 4.** a) Please see our remarks above. b) The choice of $p$ reflects the gap between the worst case in theory and
what happens in practice: the best singleton is expected to be small and non-monotone objectives are not expected to
be—loosely speaking—arbitrarily far from monotonicity. If one was to parameterize, not only by $\max_i v(i)$ (Thm. 3),
but also by $\min_{i,T} v(i \mid T)$, then it would be clear that as the former decreases and the latter increases, the optimal value
of $p$ gets closer to 1. While it is easy to construct *synthetic* instances where $p = 0.41$ and SAMPLEGREEDY performs
better than FANTOM and *arbitrarily* better than GREEDY, we felt that this would be a biased comparison (in our favor).

[Meta-Review · NeurIPS 2020]

This paper considers a new problem of optimizing a submodular function subject to a constraint. The algorithmic techniques following mostly from prior work, but the paper addresses a problem of interest that somehow missed being addressed by the community. There is some novelty in the algorithmic and analysis approach that the reviewers appreciated.